# A Prospective Study of the Serological, Clinical, and Epidemiological Features of a SARS-CoV-2 Positive Pediatric Cohort

**DOI:** 10.3390/children9050665

**Published:** 2022-05-05

**Authors:** Ignacio Callejas-Caballero, Alba Ruedas-López, Arantxa Berzosa-Sánchez, Marta Illán-Ramos, Belén Joyanes-Abancens, Andrés Bodas-Pinedo, Sara Guillén-Martín, Beatriz Soto-Sánchez, Isabel García-Bermejo, David Molina-Arana, Juan-Ignacio Alós, Elvira Baos-Muñoz, Alberto Delgado-Iribarren, Manuel E. Fuentes-Ferrer, José T. Ramos-Amador

**Affiliations:** 1Department of Paediatrics, Collaborator at Instituto de Investigación Sanitaria del Hospital Universitario Clínico San Carlos (IdISSC), 28040 Madrid, Spain; icallejascab@gmail.com (I.C.-C.); aranire@msn.com (A.B.-S.); marta_illan_ramos@hotmail.com (M.I.-R.); bjoyanes@ucm.es (B.J.-A.); 2Department of Microbiology, Collaborator at Instituto de Investigación Sanitaria del Hospital Universitario Clínico San Carlos (IdISSC), 28040 Madrid, Spain; alba.ruedas.lo@gmail.com (A.R.-L.); elvira.baos@salud.madrid.org (E.B.-M.); adelgadoi@salud.madrid.org (A.D.-I.); 3Department of Paediatrics, Hospital Universitario de Getafe, 28905 Madrid, Spain; sguillenmartin@hotmail.com (S.G.-M.); beasoto80@hotmail.com (B.S.-S.); 4Department of Microbiology, Hospital Universitario de Getafe, 28905 Madrid, Spain; igarcia.hugf@salud.madrid.org (I.G.-B.); dmolinaa@salud.madrid.org (D.M.-A.); nachoalos@telefonica.net (J.-I.A.); 5Statistician at Hospital Universitario Clínico San Carlos, 28040 Madrid, Spain; mfuentesferrer@gmail.com; 6Department of Paediatrics, Universidad Complutense-Instituto de Investigación Sanitaria del Hospital Universitario Clínico San Carlos (IdISSC), 28040 Madrid, Spain; josetora@ucm.es

**Keywords:** RT-PCR (reverse transcriptase polymerase chain reaction), SARS-CoV-2, COVID-19, seroconversion, children

## Abstract

Background: SARS-CoV-2 was a global pandemic. Children develop a mild disease and may have a different rate of seroconversion compared to adults. The objective was to determine the number of seronegative patients in a pediatric cohort. We also reviewed the clinical–epidemiological features associated with seroconversion. Methods: A multicenter prospective observational study during September–November 2020, of COVID-19, confirmed by reverse transcription-polymerase chain reaction. Data were obtained 4–8 weeks after diagnosis. Blood samples were collected to investigate the humoral response, using three different serological methods. Results: A total of 111 patients were included (98 symptomatic), 8 were admitted to hospital, none required an Intensive Care Unit visit. Median age: 88 months (IQR: 24–149). Median time between diagnosis and serological test: 37 days (IQR: 34–44). A total of 19 patients were non-seroconverters when using three serological techniques (17.1%; 95% CI: 10.6–25.4); most were aged 2–10 years (35%, *p* < 0.05). Univariate analysis yielded a lower rate of seroconversion when COVID-19 confirmation was not present amongst household contacts (51.7%; *p* < 0.05). Conclusions: There was a high proportion of non-seroconverters. This is more commonly encountered in childhood than in adults. Most seronegative patients were in the group aged 2–10 years, and when COVID-19 was not documented in household contacts. Most developed a mild disease. Frequently, children were not the index case within the family.

## 1. Introduction

The disease caused by SARS-CoV-2 (Severe Acute Respiratory Coronavirus 2) was declared a global pandemic by the World Health Organization (WHO, Geneva, Switzerland) in March 2020, and it was named COVID-19 (Coronavirus disease 2019). Although in all of the pandemic waves, the pediatric population have developed a mild disease [1,2,3], a small proportion of children have had severe clinical outcomes [4], or even a multisystem inflammatory syndrome, SARS-CoV-2 (MIS-C) [5]. There is controversy regarding whether children are more or less prone to infection than adults. While some studies have shown that the risk of infection is similar [6], others indicate that is lower [4,7,8], particularly at an early age [9]. In adulthood, non-seroconverters are mainly pauci-symptomatic patients or present with a very mild form of the disease [10,11,12], and higher titers of antibodies are found in more severe cases [13]. Children tend to develop mild or even asymptomatic forms of the disease more often than adults [14,15], and therefore, the number of patients in which no antibodies are detected might be higher in childhood when compared to adults [16,17,18]. There is evidence that the level of antibodies against the receptor-binding domain (RBD) of SARS-CoV-2’s Spike protein correlates with neutralizing antibodies, and these are of major importance for protection against future infections. However, the kinetics of these two types of antibodies might not be the same [19]. Little is known about the evolution of antibodies against different antigens of SARS-CoV-2 in children, and until now, we do not have much available information [20,21]. Knowledge of the duration and kinetics of antibodies against COVID-19 in children could help with understanding the response to vaccination, and the possibility of reinfection. 

The main objectives of this study were to analyze the proportion of seronegative patients within 4–8 weeks after infection, using three different serological methods and the concordance between them, and to describe the possible features associated with non-seroconversion. The study was performed in a pediatric population treated in two university hospitals in Madrid during the second wave of the COVID pandemic in Spain. Other aims were to describe the clinical and epidemiological aspects, and the route of infection amongst family members. 

## 2. Materials and Methods

### 2.1. Study Design

A multicenter, prospective, and observational study was conducted at two university hospitals in Madrid, (Hospital Clínico San Carlos and Hospital de Getafe), between September and November 2020, during the second wave of the COVID-19 pandemic in Spain. At that time, in order to increase the detection rate of SARS-CoV-2, and to improve the tracking of contacts, a RT-PCR test was performed for all patients with symptoms that were related to COVID-19 who were attending the Pediatric Emergency Department. Those symptoms included any of the following: fever, respiratory distress, or gastrointestinal or skin symptoms, as well as a history of close contact with a patient diagnosed with COVID-19 [22]. 

Inclusion criteria: Children and adolescents aged 0–18 that were seen in the emergency room with confirmed SARS-CoV-2 infection were included. Confirmed infection was defined when RT-PCR (reverse transcription-polymerase chain reaction) in a sample obtained from a nasopharyngeal swab was positive. Those with the confirmed infection were asked to participate in the present study, and were referred as outpatients to the clinic 4–8 weeks later. They had to fulfill a clinical–epidemiological questionnaire noting all pre- and post-symptoms after visiting the Pediatric Emergency Department. A blood sample was taken for serological tests at that point. RT-PCR was not repeated. The exclusion criteria were: immunosuppression, refusal to sign the informed consent, extraction of the blood sample for serology after the period established (4–8 weeks after the positive RT-PCR result), and inability to perform the 3 serological tests due to an insufficient serum sample.

### 2.2. Data Collection and Study Variables

The following data were obtained: serological status (a seropositive patient was defined as possessing the presence of the humoral response in at least one of the 3 serological tests, and a seronegative patient if the humoral response was not detected in any of the tests 4 to 8 weeks after diagnosis), demographic features (age, sex, date of birth, place of birth, parental origin), past medical history, clinical manifestations described in medical reports at the Emergency Room, and information from a questionnaire that the patients filled during the first outpatient visit 4 to 8 weeks after diagnosis. Clinical manifestations were: fever (≥38 °C), cough, dyspnea, gastrointestinal symptoms (abdominal pain, nausea, vomiting, and/or diarrhea), skin lesions, neurological symptoms, and others. Data related to the diagnosis, treatments, procedures, and outcomes were also obtained at the time of diagnosis and during the follow-up period. The questionnaire also included information about whether the patient was the index case or not. The index case was defined as the subject with a confirmatory RT-PCR for SARS-CoV-2 infection, with an earlier onset date of symptoms in a specific setting. Cases with a symptoms onset of less than 24 h from the index case were considered as co-primary cases. The subject in contact with an index case with a positive diagnostic test that was 24 h or more after the date of the positive test of the primary or co-primary case was defined as a secondary case. When symptoms developed 24 or more hours after the initiation of symptoms of the primary or co-primary case, it was also defined as a secondary case [23].

The study was conducted in agreement with the Declaration of Helsinki, and approved by the Ethics Committee of the Hospital Clínico San Carlos (20/647-E_COVID, 19 October 2020). An informed consent signed by the parents or the legal guardians was required in all patients, as well as informed assent by mature minors (over 12 years old) following current regulations (Declaration of Helsinki, Law 14/2007 of July 3 on Biomedical Research). 

### 2.3. Microbiological Tests

RT-PCR analysis of the nasopharyngeal swab was used for the detection of SARS-CoV-2. The detection of serum antibodies against SARS-CoV-2 was carried out according to the manufacturer’s instructions for 3 serological tests: Siemens (ADVIA Centaur ^®^ XP SARS-CoV-2 Total (COV2T)); Abbott (Alinity^®^ SARS-CoV-2 IgG II); and the anti-IgG/A/M SARS-CoV-2 ELISA test (Human IgG/IgA/IgM anti-SARS-CoV-2 ELISA by The Binding Site Group Ltd., Birmingham, UK). The COV2T test (Siemens) is a chemiluminescent immunoassay (CLIA) for the qualitative detection of IgG and IgM antibodies against the spike protein (IgG-S and IgM-S) of SARS-CoV-2 in serum and plasma. This test uses an RBD antigen contained in the S1 subunit of the SARS-CoV-2 spike protein (S). The results are considered positive if index is ≥1 and negative if <1. The Abbott SARS-CoV-2 IgG II assay is a chemiluminescent microparticle immunoassay (CMIA). This test performs a qualitative determination of IgG antibodies against the nucleocapsid protein (N) of SARS-CoV-2 (IgG-N) in serum and plasma. An index result ≥1.4 is considered positive, and negative when it is <1.4. Both chemiluminescent techniques (Abbott and Siemens assays) measure the chemiluminiscent reaction as relative light units (RLU). The index is then calculated by the systems via comparison of the chemiluminescent RLU in the reaction, to the calibrators. The Anti-SARS-CoV-2 IgG/IgA/IgM assay is based on the determination of anti-RBD IgG/IgA/IgM antibody titers using the ELISA technique (enzyme-linked immunosorbent assay or immunosorbent assay linked to enzymes). Antibody titers are estimated via the generation of isotype-specific standard curves, using monoclonal anti-SARS-CoV-2 IgG, IgA, and/or IgM antibodies. Following the manufacturer’s instructions, positive samples were identified as those with a UR/mL that was three standard deviations above the negative control samples mean.

### 2.4. Statistic Analysis

Statistical analysis was performed with STATA 15.0. The qualitative variables were presented with their absolute and relative frequency distributions. The quantitative variables were summarized with median and IQR. Statistical analysis between the qualitative variables with seronegativity at 4–8 weeks was evaluated using the chi-squared test or Fisher’s exact test. The overall agreement between the three tests was studied via the calculation of the kappa index, together with its 95% confidence interval. A significance value of 5% was accepted for all tests. Data processing and analysis was carried out using the statistical package STATA v.15.0.

## 3. Results

Amongst all of the consecutive patients who attended the Pediatric Emergency Department in both hospitals, 144 were selected. The following were excluded: 12 because the blood sample was not obtained 4 to 8 weeks after the diagnosis of COVID-19, 2 because of immunosuppression (human immunodeficiency virus, juvenile rheumatoid arthritis treated with rituximab), 19 due to insufficient blood samples to perform the three serological tests. Eventually, 111 patients were included.

Regarding serology, a seropositive patient was defined as having the presence of the humoral response in at least one of the three serological tests, and a seronegative patient if the humoral response was not detected in any of them. Abbott´s test for the detection of IgG-N was negative in 25 (22.5%) cases, compared to 22 (19.8%) when using the Siemens test for the detection of IgG-S and IgM-S, and 20 (18%) when using the Binding Site test that detects anti-RBD IgG/IgA/IgM antibodies. The median time for obtaining blood samples for serology from the time of positive diagnosis via RT-PCR was 37 days (IQR: 33–44). Global agreement between the three techniques, as measured via the kappa index, was high (kappa: 0.89; 95% CI: 0.76–0.95), presenting an absolute agreement of 94.5%. All three methods were coincident for a positive result in 86 cases (77.5%), and for a negative result in 19 cases (17.1%). When comparing the tests two by two, the results were as follows: SARS-CoV-2 IgG II, Abbott with COV2T, Siemens (kappa: 0.92; IC 95%: 0.78–0.97); SARS-CoV-2 IgG II, Abbott with Human IgG/IgA/IgM anti-SARS-CoV-2 ELISA by The Binding Site Group (kappa: 0.86; IC 95%: 0.69–0.94), and COV2T, Siemens with Human IgG/IgA/IgM anti-SARS-CoV-2 ELISA by The Binding Site Group (kappa: 0.89; IC 95%: 0.72–0.95). The number of patients with seronegative results 4 to 8 weeks after the microbiological diagnosis via RT-PCR was 19 (17.1% (95% CI: 10.6–25.4)). Table 1 shows the relationships between the different clinical and epidemiological variables with seroconversion. Features that showed a statistically significant association with a lower seroconversion were an age range of 2–10 years (16 cases; 35.5%, 95% CI 21.6–49.5) and a history of disease without confirmation in household contacts (15 cases, 51.7%, 95% CI 33.5–69.9). 

Patients symptomatology is shown in Table 2. Only eight patients (7.2%) required hospital admission, with a median stay of 2 days (IQR: 2–4), and none of them in PICU. Most required only hospital monitoring because of their young age, one required oxygen therapy, and two required intravenous fluid therapy due to low oral intake. None of them required corticosteroids, other immunomodulators, or antiviral therapies. The patient was the index case within the family in 53 cases (47.7%). In 58 cases (53.3%), the index case patient was generally a family member (96.4%). In most children (88 cases; 79.2%), household contacts had symptoms that were consistent with COVID-19, and 81 (73%) were confirmed as positive via an RT-PCR test. There was no confirmation of infection amongst household contacts in 30 cases (27%).

## 4. Discussion

The COVID-19 infection gives rise to a humoral or cellular immune response in most cases, but the duration of this immunity and its protection against reinfection remains unclear [24]. Concerning the humoral response seen in our study, we found a high proportion of non-seroconverters using three different serological methods from a single blood sample 4 to 8 weeks after a diagnosis of COVID-19. A high correlation among the three tests used was found. The detection of antibodies using ELISA human IgG/IgA/IgM anti-SARS-CoV-2 yielded a higher sensitivity than with the Siemens and Abbott techniques (18% seronegativity vs. 19.8% and 22.5%, respectively), although the number of patients was small and comparisons were therefore not appropriate. It is possible that the variability in the kinetics of the various antibodies accounted for the slight difference in sensitivity between the three types of serological tests. The concordance between the three techniques after the statistical analysis between them reinforces our results, and might be related to the high grade of seronegativity found in childhood compared to adulthood [11]. In this respect, these results have called our attention to compare them with what has been previously published in adult patients, as perhaps this is a more common situation in the pediatric age group than previously thought [11]. The fact that infection in childhood is frequently asymptomatic or pauci-symptomatic may explain why the proportion of individuals in whom no antibodies are detected is higher amongst the pediatric population compared to adult patients. Within this likely higher seronegativity found in children, in our series, there is a statistically significant association between no seroconversion and the ages 2–10, as well as unconfirmed COVID-19 in co-habitants. By contrast, the presence of some symptoms, such as neurological involvement, was associated with seroconversion. There was also a statistical association if seroconversion was confirmed as being COVID-19 in the family members. Although the significance of these findings is uncertain, it could be related to the mild symptomatology that is commonly found in this age bracket, or to the lower rate of exposure to the virus outside the family group. 

In addition, our study confirms a mild disease course in childhood during the second wave of the SARS-CoV-2 pandemic in Spain [2,25,26]. According to other pediatric series, the most commonly reported symptoms were fever, followed by mild respiratory symptoms, headache, and gastrointestinal disease [6]. During the second wave of the pandemic, RT-PCR was performed on a nasopharyngeal sample as a protocol for every child with symptoms that were consistent with COVID-19 that attended the emergency room. This might explain why in our series, a mild course of the disease was more common compared to the first-wave studies, where the test was rarely performed [26]. As previously reported, the index case at home was a child for less than 50% of the cases, indicating therefore that adults are the main source of infection [7]. In our study, a large proportion of children were living with adults with symptomatic COVID-19 or were confirmed via RT-PCR as having the disease. Although there is the possibility of false-positive RT-PCR results in some of our patients who did not seroconvert, we think that this is quite unlikely, given the high specificity of the technique and the fact that almost all of them were symptomatic during a high incidence period in our setting [27,28].

This study has several limitations. The number of patients included was relatively small; however, patients were recruited consecutively during the study period. In addition, this was a prospective analysis that lacked the obtained retrospective data of a clinical and epidemiological questionnaire; therefore, the memory bias of both the parents and children may have been present. Data were collected via questionnaires in an outpatient facility, and some of them were not obtained at the right moment when the patient attended the Emergency Room. In some cases, we did not have accurate information about some of the symptoms, such as anosmia. External validity of our study is limited because it only includes patients that were treated in an emergency setting, and does not necessarily reflect what happens in the general population. As previously mentioned, we cannot completely rule out the possibility of false positives using RT-PCR analysis of nasopharyngeal swabs, or that contact patients with a clinical suspicion of COVID-19 had other diseases. Nevertheless, our study has several strengths. On the one hand, it is a prospective design that was conducted in the second wave of the pandemic in Spain, with a selection according to the pre-established criteria of consecutive patients with a RT-PCR-confirmed diagnosis. On the other hand, the serological tests were performed in the same time lapse after the infection, using different laboratory methods. 

In summary, our results offer additional data suggesting that the proportion of children in whom seroconversion is not detected may be higher than in adults. Furthermore, our data confirm a mild clinical course in the majority of children, and that they are usually not the index case in the family. More studies are needed to determine the possible factors involved in the humoral response and its impact in SARS-CoV-2-infected children.

## Figures and Tables

**Table 1 children-09-00665-t001:** Factors associated with seronegativity and seropositivity.

Patient Characteristics	Total Population (111) *n*%	Seronegative *n*%	Seropositive *n*%	*p* Value
Previous pathology	31 (27.9%)	9 (8.1%)	22 (19.8%)	0.038
Age <2 years 2–10 years >10 years	26 (23.4%)	2 (1.8%)	24 (21.6%)	<0.001
	45 (40.5%)	16 (14.4%)	29 (26.1%)	
	40 (36.0%)	1 (0.9%)	39 (35.1%)	
Gender Male Female	53 (47.7%)	11 (9.9%)	42 (37.8%)	0.331
	58 (52.2%)	8 (7.2%)	50 (45.0%)	
Parents origin *Spain* *Central and South America* Other	49 (44.1%)	11 (9.9%)	38 (34.2%)	0.136
	31 (27.9%)	2 (1.8%)	29 (26.1%)	
	31 (27.9%)	6 (5.4%)	25 (22.5%)	
Close contacts study	17 (15.3%)	3 (2.7%)	14 (12.6%)	0.950
Asymptomatic patient	13 (11.7%)	2 (1.8%)	11 (9.9%)	0.860
Fever: Low grade fever	78 (70.2%)	13 (11.7%)	65 (58.5%)	0.846
Respiratory symptoms	61 (85%)	13 (11.7%)	48 (43.2%)	0.195
Gastrointestinal symptoms	42 (37.8%)	5 (4.5%)	37 (33.3%)	0.255
Neurological symptoms	41 (36.9%)	3 (2.7%)	38 (34.2%)	0.036
Cutaneous symptoms	8 (7.2%)	2 (1.8%)	6 (5.4%)	0.539
General symptoms	45 (40.5%)	5 (4.5%)	40 (36.0%)	0.155
Hospitalization	8 (7.2%)	0 (0.0%)	8 (7.25)	0.182
Index case patient	53 (47.7%)	16 (14%)	37 (33.3%)	0.001
COVID-19 confirmation in household contacts	81 (73%)	4 (3.6%)	77 (69.3%)	<0.001

**Table 2 children-09-00665-t002:** Patient symptoms.

Patient Symptoms	
Symptomatology	*n* (%)
Present complaint	
Clinical suspicion	98 (88.2)
Asymptomatic	13 (11.7)
Fever: Low grade fever	78 (70)
Respiratory	61 (55.0)
Rhinorrhea	49 (44.1)
Cough	35 (31.5)
Shortness of breath	10 (9)
Chest pain	2 (1.8)
Gastrointestinal	42 (37.8)
Diarrhea	22 (19.8)
Abdominal pain	19 (17,1)
Vomiting	16 (14.4)
Neurological	41 (36.9)
Headache	33 (29.7)
Anosmia	18 (16.2)
days *	12 (6–30)
Ageusia/dysgeusia	15 (13.5)
days *	7 (4–10)
Dizziness	2 (1.8)
Cutaneous	8 (7.2)
Nonspecific rash	3 (2.7)
Eczema	2 (1.8)
Petechiae	1 (0.9)
Urticaria	1 (0.9)
General symptoms	45 (40.9)
Hyporexia	14 (12.6)
Odynophagia	8 (7.2)
Irritability	4 (3.6)

* Data expressed as median and interquartile range.

## Data Availability

The data presented in this study are available on request from the corresponding author. The data are not publicly available due to patient confidentiality.

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
