# Peer review of "A Prospective Study of the Serological, Clinical, and Epidemiological Features of a SARS-CoV-2 Positive Pediatric Cohort"

_children, 2022, doi:10.3390/children9050665_

Round 1

Reviewer 1 Report

Thank you dear authors. The subject selection is quite up-to-date and interesting. The design of the study is also simple and applicable. However, the result and discussion sections need to be revised. The result section is very complex and difficult to understand. It is necessary to shorten the sentences, especially serological findings should be brought to the fore. Because the main purpose of the article was not to indicate the clinical and demographic characteristics of covid patients. In the discussion part, they were required to discuss their own findings more heavily. 

Author Response

Dear reviewer, thank you very much for your suggestions.

 We agree to simplify the result section for better understanding since so much data can be confusing.

We have therefore decided to remove the clinical and demographic characteristics from the text and keep it only in the tables to highlight the serological findings, which is the main objective of the study

In the discussion section, as suggested, we further emphasize the importance of the non-seroconversion found in a large percentage of patients by 3 different serological techniques, which we believe is the most relevant part of our study compared to what has been published to date in articles on the adult population (discussion section, paragraphs 1-2)

Reviewer 2 Report

This manuscript addresses the issue that children may differ from adults in their immune-response to SARS-CoV-2. The authors have a very interesting population at hand and a almost unique setup in that all these children from the early phase of the pandemic had an RT-PCR done centrally and early as well as late symptoms were documented prospectively. However, a number of points in the manuscript can be improved.

Major

  1. Overall and specifically in the methods section, a paragraph based writing style needs to be implemented. One sentence paragraphs are not acceptable.
  2. When the authors write that little is known about the topic in the introduction, they cite reference [20]. There also Laub et al, Front. In Pediatrics 2021 must be cited. That paper has a very similar topic as the paper here.
  3. The study design section in the methods part needs to be more systematic. When was it performed and where? What were inclusion and exclusion criteria? How was the flow of the tests and assessments. Use paragraphs!
  4. The RT-PCR method needs to be reported in detail. I assume that this was done centrally in one (or maximum 2 laboratories) given the setup. One possible explanation for the low seroconversion rate is that RT-PCR results are wrong. Especially, when no other family member was positive. This has to be addressed and should be excluded, giving the QC measures for PCR. The central PCR is a strength of the study!
  5. A table comparing the three serological tests would be helpful: what is each test detecting? What is the specificity of each test? What is sensitivity? Value range? Cut off? Reference in the literature with outer populations. Range and median values for adult population with that test if available in the literature.
  6. Results: please give the definition of what is seronegative. I would suggest that this is negative in all three tests.
  7. A correlation graphic for the three serological tests would be very much appreciated.
  8. Results: Instead of table 1 (which is totally confusing) and table 2 I would urgently suggest to use one table in the following layout (and maybe the important issues from table 33 can also go into this congregated table):

Describe the factors on the left side, give values and % for: total population (N= 111), seronegative (n=19) and seropositive (n=92), give p-value for differences between seronegative and positive on far right side.

  1. Results: what do the symptoms given in table 3 relate to? Acute symptoms or symptoms at the second visit? I do not see how that relates to the main topic of the manuscript but if it does in the eyes of the authors, they should explain it better (or leave it out otherwise).
  2. Discussion: in the second paragraph of the discussion, which is a prominent spot in a discussion, the authors discuss the durability of the antigen response (citation [18]). However, they do not show data on that in their own work. I would suggest to either leave it out or move it to a less prominent spot to avoid confusion of the reader. Again, do not use one sentence paragraphs.
  3. The possibility that the RT-PCR may have been false-positive has to be addressed. It is also unclear, if RT PCR were performed in those families in which you detected positive children first.

Minor

  1. Abstract, sentence 2 unfinished: replace Emergencies with Emergency departments.
  2. The sentence” Mild clinical picture..” at the very end is out of context and unnecessary for the abstract and distracts from the main message.
  3. Methods: I am not so sure if you should call RT-PCR and serological Tests “microbiological tests as in the header of section 2.3. Laboratory tests is more adequate here. RT-PCR needs to be explained in a paragraph here.
  4. Methods 2.4: statistical analysis was performed, skip “out”

Author Response

Dear reviewer, thank you very much for dedicating your time to correcting and improving the results of our study in Spain.

We have taken into account all your suggestions, as detailed below.

Also, the manuscript has been reviewed by an English language expert as requested. English language and style have been reviewed by an English language expert

  1. Thank you very much for indicating us the need to implemenet a paragraph based writing style for a better reading of the article. We have fixed that aspect in this new versión.
  2. Thank you very much, it is a very interesting article that we have added in the text with the reference number 20 (introductory section, paragraph 1). We have changed numbers in subsequent bibliography as appropriate.

  1. We appreciate the comment that the study design section was somewhat “confusing” and we have tried to do it more systematic. We have modified it based on suggestions for better understanding.

  1. As mentioned, there is a possibility of false-positive results in the RT-PCR method and that this may have resulted in seronegativity. According to the Microbiology Service of both hospitals, the RT-PCR method has been shown to have high specificity, so we do not believe this could have occurred.

In addition, we must take into account the high positive predictive value of the technique in symptomatic patients and the epidemic period. The technique was performed in the two hospitals mentioned in the article (Clínico San Carlos and Getafe), as it was impossible to centralize it in a single hospital since the samples had to be sent immediately to the Microbiology Service for processing. Both hospitals are two important tertiary centers in the Community of Madrid with a high volume of patients and a strong Microbiology Service that has been in operation for many years, always working with the same methodology. Attached in the e-mail are available the quality standards of both hospitals provided by the Microbiology Services and guaranteed by the Spanish Society of Microbiology (SEIMC)

We also think it is appropriate to comment on this possibility for the reader and to include a bibliography in the article reinforcing the low possibility of false positives (discussion section, paragraphs 2-3). We, therefore, thank you very much for your comment.

  1. We appreciate the suggestion. Following the recommendation, we have tried to clarify this section by creating a summary table with the data showing the different techniques.

This table below (Table 1 in this document) can be included in the document if you consider it. We are aware that there is a lot of technical information in the methods section that can be difficult to understand on first reading. However, we feel that detailing each technique in greater depth would provide too much information that can easily be found in the manufacturer's package insert, if readers are interested.The manufacturer’s inserts are available and could be sent or provided as supplementary content if you consider.

Regarding the sensitivities of the tests, it is difficult to give a precise answer because a negative result may be because some patients do not seroconvert or to a false negative of the technique. In adults, for example, some studies report sensitivities ranging from 85,7%-100% for the Abbott test when the assay is performed at least more than 14 days after the onset of symptoms. (references A-C)

  1. A. Nicol T, Lefeuvre C, Serri O, Pivert A, Joubaud F, Dubée V, et al. Assessment of SARS-CoV-2 serological tests for the diagnosis of COVID-19 through the evaluation of three immunoassays: Two automated immunoassays (Euroimmun and Abbott) and one rapid lateral flow immunoassay (NG Biotech). J Clin Virol. 2020;129:104511.
  2. B. Manalac J, Yee J, Calayag K, Nguyen L, Patel PM, Zhou D, et al. Evaluation of Abbott anti-SARS-CoV-2 CMIA IgG and Euroimmun ELISA IgG/IgA assays in a clinical lab. Clin Chim Acta. 2020;510:687-90.
  3. C. Jugwanth S, Gededzha MP, Mampeule N, Zwane N, David A, Burgers WA, et al. Performance of the Abbott SARS-CoV-2 IgG serological assay in South African 2 patients. PLoS One. 2022;17(2):e0262442.

Table 1.  Comparative of the three serological tests

SARS-CoV-2 Total

(Siemens)

SARS-CoV-2 IgG II

(Abbott)

Human IgG/IgA/IgM anti-SARS-CoV-2

(The Binding Site Group Ltd.)

Serological technique

CLIA

CMIA

ELISA

Antibodies detected

Anti-RBD IgG and IgM

Anti-Nucelocapsid IgG

Anti-RBD IgA, IgG and IgM

Units

RLU

RLU

UR/mL

Value range

0,05-10,00 (Index)

c

0.774-1.132

Cut off

≥1

≥1.4

≥1.0

Specificityª

100%

99,1%

99.3%

Sensitivity b

89/111 (80,2%)

86/111 (77,5%)

91/111 (82,0%)

a Calculated from manufacturer´s insert.

b Compared with RT-PCR.

c Exact value is not shown in manufacturer´s insert.

  1. Thank you for the suggestion. The definition of seronegative was previously described in the study design section (point 2.2 Data collection and study variables, paragraph 1). We agree with you that a seropositive patient is defined as the presence of humoral response in at least one of the 3 serological tests, and a seronegative patient if the humoral response is not detected in any of the three tests 4 to 8 weeks after diagnosis. To clarify the Reading, we have also added this definition in the result section (results section, paragraph 2)

  1. Thank you for your It is suggested to include a graphic with te correlation for the three serological techniques. We had included in the text the Kappa index showing a very good concordance among the three different techniques (results section, paragraph 2). You can see in the following tables (Table 2 and Table 3 in this document) the detailed data of the serological techniques that we may incorporate if you consider necessary.

Table 2. Serological techniques and results

Serological technique

Seropositive

Seronegative

Human IgG/IgA/IgM anti-SARS-CoV-2

(The Binding Site Group Ltd.)

91/111 (82,0%)

20/111 (18,0%)

SARS-CoV-2 Total

(Siemens)

89/111 (80,2%)

22/111 (19,8%)

SARS-CoV-2 IgG II

(Abbott)

86/111 (77,5%)

25/111 (22,5%)

3 techniques coincident

86/111 (77,5%)

19/111 (17,1%)

Table 3. Serological techniques correlation

Compared techniques

Kappa index

Confidence Interval 95%

Human IgG/IgA/IgM anti-SARS-CoV-2

(The Binding Site Group Ltd.)

Vs

SARS-CoV-2 IgG II

(Abbott)

0,86

0,69-0,94

Human IgG/IgA/IgM anti-SARS-CoV-2

(The Binding Site Group Ltd.)

Vs

SARS-CoV-2 Total

(Siemens)

0,89

0,72-0,95

SARS-CoV-2 Total

(Siemens)

Vs

SARS-CoV-2 IgG II

(Abbott)

0,92

0,78-0,97

  1. We have taken into account the suggestion by merging and simplifying tables 1 and 2 into a new table 1 in the article (table 4 in this document) since some sections were repeated (age, gender and origin).

As for the origin, we have decided, if you agree, to highlight only the simplified origin of the parents (Spanish, Central and South America and others) and to remove the place of birth of the children since this data is not particularly relevant for our study. After discussing it again with our team and with the statistical expert (Manuel Fuentes Ferrer), we have considered to conform the table accordingly, so that the serological status within each relevant variable is shown.

In this regard, we believe it is appropriate to keep the table of symptoms separately (new table 2), since the statistical comparison has been made by groups of symptoms.

We have noticed that some data were missed unwillingly although the final results do not change:

-The asymptomatic patients were 13 and not 17 (data previously confused in the table with close contact study).

- In the section on cutaneous manifestations, we were missing a patient with

    nonspecific rash to add up the 8 total cases with cutaneous involvement.

- Within the general symptoms, we have added the missing patients with 

  myalgias/arthralgias and asthenia.

As you may see in the table 4, the presence of personal history and neurological symptoms (anosmia and dysgeusia, among others), also have statistical significance. For this reason, as we consider the presence of neurological symptomas as a relevant finding, we have included a comment in the article (discussion section, paragraph 1).

Table 4. Factors associated with seronegativity and seropositivity

Patient characteristics

Total population (111)

n%

Seronegative n%

Seropositive n%

P value

Previous pathology

31 (27,9%)

9 (8,1%)

22 (19,8%)

0.038

Age

<2 years

2-10 years

>10 years

26 (23,4%)

2 (1,8%)

24 (21,6%)

< 0.001

45 (40,5%)

16 (14,4%)

29 (26,1%)

40 (36,0%)

1 (0,9%)

39 (35,1%)

Gender

Male

Female

53 (47,7%)

11 (9,9%)

42 (37,8%)

0.331

58 (52,2%)

8 (7,2%)

50 (45,0%)

Parents origin

Spain

Central and SoutAmerica

Other

49 (44,1%)

11 (9,9%)

38 (34,2%)

0.136

31 (27,9%)

2 (1,8%)

29 (26,1%)

31 (27,9%)

6 (5,4%)

25 (22,5%)

Close contacts study

17 (15,3%)

3 (2,7%)

14 (12,6%)

0.950

Asymptomatic patient

13 (11,7%)

2 (1,8%)

11 (9,9%)

0.860

Fever-Low grade fever

78 (70,2%)

13 (11,7%)

65 (58,5%)

0.846

Respiratory symptoms

61 (85,0%)

13 (11,7%)

48 (43,2%)

0.195

Gastro intestinal symptoms

42 (37,8%)

5 (4,5%)

37 (33,3%)

0.255

Neurologycal symptoms

41 (36,9%)

3 (2,7%)

38 (34,2%)

0.036

Cutaneous symptoms

8 (7,2%)

2 (1,8%)

6 (5,4%)

0.539

General symptoms

45 (40,5%)

5 (4,5%)

40 (36,0%)

0.155

Hospitalization

8 (7,2%)

0 (0,0%)

8 (7,25)

0.182

Index case patient

53 (47,7%)

16 (14,4%)

37 (33,3%)

0.001

COVID-19 confirmation in household contacts

81 (73,0%)

4 (3,6%)

77 (69,3%)

<0.001

  1. The symptoms in Table 3 (Table 2 in the new version) referred to those that occurred from the onset of symptoms until the visit 4-8 weeks after diagnosis. Perhaps this data, which has been further specified in the article (study design section, paragraph 2), was not well reflected initially. Indeed, it is not a main objective of the study but we believe that it may be of interest to the reader, so we have kept these data in Table 2 of the new versión of the article, but have removed it from the text.

  1. Thank you for the suggestion. Indeed, the section on durability antibody response is slightly related but it is not the main topic of this study and no data on the durability of antibodies is reported. We have shortened this aspect in the introduction section (paragraph 1) and removed it from the results section in order to focus on the really important points of our study, which are the prevalence of seropositivity and factors associated with no seroconversion.

  1. The possibility that the RT-PCR may have been false-positive has been adressed (discussion section, paragraphs 2-3) as mentioned in section 4 above.

Regarding the performance of PCR on family members, the local health policy at that time recommended performing PCR on the cohabitants of cases diagnosed with COVID-19 for correct case notification and this was attempted in most patients. However, this work depended mainly on the outpatient Primary Care Health Service and not so much on the Hospital setting. As a result we do not have further information than that provided in the article.
